# Real-World Therapy with Pembrolizumab: Outcomes and Surrogate Endpoints for Predicting Survival in Advanced Melanoma Patients in Germany

**DOI:** 10.3390/cancers14071804

**Published:** 2022-04-01

**Authors:** Peter Mohr, Emilie Scherrer, Chalid Assaf, Marc Bender, Carola Berking, Sheenu Chandwani, Thomas Eigentler, Imke Grimmelmann, Ralf Gutzmer, Sebastian Haferkamp, Jessica C. Hassel, Axel Hauschild, Rudolf Herbst, Ruixuan Jiang, Katharina C. Kähler, Clemens Krepler, Alexander Kreuter, Ulrike Leiter, Carmen Loquai, Friedegund Meier, Claudia Pföhler, Anja Rudolph, Dirk Schadendorf, Maximo Schiavone, Gaston Schley, Patrick Terheyden, Selma Ugurel, Jens Ulrich, Jochen Utikal, Carsten Weishaupt, Julia Welzel, Michael Weichenthal

**Affiliations:** 1Department of Dermatology, Elbe Kliniken Buxtehude, 21614 Buxtehude, Germany; peter.mohr@elbekliniken.de (P.M.); marc.bender@elbekliniken.de (M.B.); 2Merck & Co., Inc., Kenilworth, NJ 07033, USA; emi.scherrer@gmail.com (E.S.); sheenuchandwani@gmail.com (S.C.); ruixuan.jiang@merck.com (R.J.); clemens.krepler@merck.com (C.K.); 3Department of Dermatology and Allergy, Skin Cancer Center Charité, Charité—Universitätsmedizin Berlin, 10117 Berlin, Germany; chalid.assaf@helios-gesundheit.de (C.A.); thomas.eigentler@uni-tuebingen.de (T.E.); 4Department of Dermatology and Venerolgy, HELIOS Klinikum Krefeld, 47805 Krefeld, Germany; 5Department of Dermatology, Universitätsklinikum Erlangen, Comprehensive Cancer Center Erlangen EMN, Deutsches Zentrum Immuntherapie, Friedrich-Alexander University Erlangen-Nuremberg, 91054 Erlangen, Germany; carola.berking@uk-erlangen.de; 6Department of Dermatology, University Hospital Tuebingen, 72076 Tuebingen, Germany; ulrike.leiter@med.uni-tuebingen.de; 7Department of Dermatology, Skin Cancer Center Hannover, Hannover Medical School, 30625 Hannover, Germany; grimmelmann.imke@mh-hannover.de; 8Department of Dermatology, Johannes Wesling Medical Center, Ruhr University Bochum, Campus Minden, 32429 Minden, Germany; ralf.gutzmer@ruhr-uni-bochum.de; 9Department of Dermatology, University of Regensburg, 93040 Regensburg, Germany; sebastian.haferkamp@klinik.uni-regensburg.de; 10Department of Dermatology and National Center for Tumor Diseases, University Hospital Heidelberg, 69120 Heidelberg, Germany; jessica.hassel@med.uni-heidelberg.de; 11Department of Dermatology, Skin Cancer Center, University Hospital Schleswig-Holstein, Campus Kiel, 24105 Kiel, Germany; ahauschild@dermatology.uni-kiel.de (A.H.); kkaehler@dermatology.uni-kiel.de (K.C.K.); 12Department of Dermatology, Helios Klinikum Erfurt, 99089 Erfurt, Germany; rudolf.herbst@helios-gesundheit.de; 13Department of Dermatology, Venereology and Allergology, HELIOS St. Elisabeth Hospital Oberhausen, University Witten/Herdecke, 46045 Oberhausen, Germany; alexander.kreuter@helios-gesundheit.de; 14Department of Dermatology, University Medical Center Mainz, 55131 Mainz, Germany; carmen.loquai@unimedizin-mainz.de; 15Skin Cancer Center at the University Cancer Centre Dresden and National Center for Tumor Diseases, 01307 Dresden, Germany; friedegund.meier@uniklinikum-dresden.de; 16Department of Dermatology, University Hospital Carl Gustav Carus, 01307 Dresden, Germany; 17Department of Dermatology, Saarland University Medical School, 66123 Homburg, Germany; claudia.pfoehler@uniklinikum-saarland.de; 18IQVIA, 60549 Frankfurt, Germany; anja.rudolph@de.imshealth.com (A.R.); maximoschiavone@gmail.com (M.S.); 19Clinic for Dermatology, Venereology and Allergy, University of Essen, 45147 Essen, Germany; dirk.schadendorf@uk-essen.de (D.S.); selma.ugurel@uk-essen.de (S.U.); 20Comprehensive Cancer Center (Westdeutsches Tumorzentrum), University Hospital Essen, Essen & German Cancer Consortium, 45147 Essen, Germany; 21Department of Dermatology, HELIOS Hospital Schwerin, 19055 Schwerin, Germany; gaston.schley@helios-gesundheit.de; 22Department of Dermatology, University of Lübeck, 23562 Lübeck, Germany; patrick.terheyden@uksh.de; 23Department of Dermatology and Allergy, Skin Cancer Center, 06484 Quedlinburg, Germany; jens.ulrich@harzklinikum.com; 24Department of Dermatology, Venereology and Allergology, University Medical Center Mannheim, Heidelberg University, 68167 Mannheim, Germany; jochen.utikal@umm.de; 25Skin Cancer Unit, German Cancer Research Center (DKFZ), 69120 Heidelberg, Germany; 26Department of Dermatology, University Hospital Münster, 48149 Münster, Germany; hauptle@uni-muenster.de; 27Department of Dermatology and Allergology, University Hospital Augsburg, 86156 Augsburg, Germany; julia.welzel@uk-augsburg.de

**Keywords:** advanced melanoma, surrogate endpoint, real-world evidence, overall survival (OS), time to next treatment (rwTtNT), pembrolizumab

## Abstract

**Simple Summary:**

Knowledge on the real-world outcomes of patients with advanced melanoma and the value of different endpoints for evaluating survival benefits is limited. We investigated the outcomes and different surrogate endpoints for overall survival (OS) in 664 pembrolizumab-treated patients with advanced melanoma in Germany. Our findings support the effectiveness of pembrolizumab in real-world clinical practice. The real-world time to next treatment was most strongly correlated with OS, suggesting it as a valuable surrogate endpoint to assess treatment effectiveness. Real-world studies assessing time to next treatment could support clinical and payer decision making.

**Abstract:**

Knowledge on the real-world characteristics and outcomes of pembrolizumab-treated advanced melanoma patients in Germany and on the value of different real-world endpoints as surrogates for overall survival (OS) is limited. A sample of 664 pembrolizumab-treated patients with advanced melanoma from the German registry ADOReg was used. We examined OS, real-world progression-free survival (rwPFS), real-world time to next treatment (rwTtNT), and real-world time on treatment (rwToT). Spearman’s rank and iterative multiple imputation (IMI)-based correlation coefficients were computed between the OS and the rwPFS, rwTtNT, and rwToT and reported for the first line of therapy and the overall sample. The median OS was 30.5 (95%CI 25.0–35.4) months, the rwPFS was 3.9 months (95%CI 3.5–4.9), the rwTtNT was 10.7 months (95%CI 9.0–12.9), and the rwToT was 6.2 months (95%CI 5.1–6.8). The rwTtNT showed the highest correlation with the OS based on the IMI (rIMI = 0.83), Spearman rank correlations (rs = 0.74), followed by the rwToT (rIMI = 0.74 and rs = 0.65) and rwPFS (rIMI = 0.69 and rs = 0.56). The estimates for the outcomes and correlations were similar for the overall sample and those in first-line therapy. The median OS was higher compared to recent real-world studies, supporting the effectiveness of pembrolizumab in regular clinical practice. The rwTtNT may be a valuable OS surrogate, considering the highest correlation was observed with the OS among the investigated real-world endpoints.

## 1. Introduction

The last decade has seen a revolution in the treatment of advanced (unresectable stage III and stage IV) melanoma with the advent of immunotherapies [1]. Immune checkpoint inhibitors, such as the programmed cell death protein 1 (PD-1) inhibitors, pembrolizumab and nivolumab, have improved the overall survival (OS) of patients with advanced melanoma compared to common chemotherapy regimens [2,3,4,5]. PD-1 inhibitors showed higher response rates and lower toxicity levels than ipilimumab [6] and are now considered a standard of care for advanced melanoma patients in the first-line setting [7,8]. Randomized clinical trials (RCTs) on pembrolizumab have yielded extensive evidence of its efficacy in advanced melanoma [4,9,10]. RCTs have high internal validity but are conducted on highly selected populations and tightly controlled care settings [11,12], making their generalizability limited with respect to real-world populations [13]. Real-world data sources, such as electronic health records (EHR) and claims databases, can provide additional meaningful insights into the outcomes of representative and heterogeneous patient populations, which contribute to a better understanding of the clinical benefits in real-world patients [14,15].

Real-world evidence (RWE) on pembrolizumab in US samples has shown median OS rates between 19.4 and 30 months [16,17,18,19] and a median real-world progression-free survival (rwPFS) of 4.2 months [19]. Recent RWE data on pembrolizumab from Slovenia showed a comparable median OS of 25.1 months, with a median rwPFS of 10.7 months [20]. Improvements in OS have been reported after the introduction of PD-1 inhibitors in Germany [21]; however, detailed insights on real-world outcomes related to pembrolizumab are still lacking.

Crucially, RWE is sourced from information collected for routine clinical practice. Therefore, challenges in data completeness and frequency of assessments to capture clinical endpoints may exist in real-world data. Research has shown that real-world effectiveness is comparable to RCT-based treatment efficacy when OS is used as an endpoint [22]. Yet, there can be gaps in real-world death data as EHRs and claims data do not routinely capture death date or cause [23]. Moreover, the timing of progression and progression-free survival may be difficult to assess precisely in the real-world setting because it can only be measured at evaluations of clinical parameters, such as imaging, biomarkers, or patient-reported outcomes, and these evaluations may occur at irregular intervals based on a range of factors, for example, patient availability [24,25]. The unavailability of the precise date of progression in structured EHRs and in tumor registries, the lack of consistent timing of tumor assessments, along with subjective assessment, pseudo-progression, and a lack of high-quality structured data are all potential limitations that may be encountered with respect to measuring progression in real-world data sources. Due to these limitations, we separately refer to real-world outcomes with the prefix “rw” to distinguish them from the stringent outcomes measured in clinical trials. Because OS can be definitively assessed, the real-world prefix is not applied.

Furthermore, rwPFS may not comprehensively reflect the outcomes with PD-1 inhibitors, where treatment beyond progression can be important for preserving tumor control and ensuring long-term efficacy [26,27]. For immunotherapies, such as pembrolizumab, proxy measures such as real-world time to next treatment (rwTtNT) and real-world time on treatment (rwToT) might, therefore, offer valuable alternative endpoints that can often be derived with higher certainty. The rwTtNT captures the interval from the initiation of a therapy to the commencement of the next line of therapy, thereby indicating the period of therapeutic benefit. rwToT captures the time interval between the index date and the date of the last dose of a therapy [28]. Evidence for the applicability of rwTtNT and rwToT to evaluate the real-world efficacy of PD-1 inhibitors in advanced melanoma patients is emerging [16]. However, more evidence for the associations between different surrogate endpoints for treatment effectiveness and OS is needed [29], including a better understanding of how time on treatment or treatment discontinuation rates correlate with OS [27]. It is, therefore, important that RWE studies systematically consider different real-world endpoints and their correlations, as recently proposed by the Friends of Cancer Research (FoCR) group [27].

ADOReg is the largest clinical registry in Germany in the field of dermatological oncology. It is maintained by the German Working Group on Dermatological Oncology (ADO), also known as the German Dermatologic Cooperative Oncology Group (DeCOG), and IQVIA. It comprises high-quality data and is, therefore, well suited for a comparative investigation of different real-world endpoints. The objectives of this study were to examine real-world clinical outcomes and to evaluate the performance of rwPFS, rwToT, and rwTtNT as surrogate endpoints of OS associated with pembrolizumab treatment in advanced melanoma patients in the German ADOReg database.

## 2. Materials and Methods

### 2.1. Study Design and Data Source

This observational retrospective study used the data of eligible patients from the skin cancer registry ADOReg of the German Dermatologic Cooperative Oncology Group [30]. The ADOReg platform was developed in 2014 and collects healthcare data on melanoma patients from 59 geographically diverse skin cancer centers or practice-based dermato-oncologists certified by the German Cancer Society (DKG), 35 of which contributed to the current study. Details on the treatment of dermatological oncology in everyday clinical practice were recorded in an unidentifiable, pseudonymized form at the patient level. The median lag time for clinical updates to the data was approximately 3 to 6 months. The ADOReg registry was approved by the ethics committee of the University Duisburg-Essen (14-5921-BO). Patient consent was obtained for inclusion in the registry, and institutional review board (IRB) approval for the ADOReg database includes the use of data for research purposes.

### 2.2. Study Population

The study included the data of adults aged ≥18 years with a confirmed diagnosis of advanced melanoma and who received ≥1 dose of pembrolizumab at the index date. The index date was defined as the date of pembrolizumab treatment initiation in a study period between 1 August 2015 and 30 June 2019. The data cut-off date was September 2019. Patients were excluded if they received pembrolizumab in a clinical trial, simultaneously received any other systemic therapy, or were treated with pembrolizumab for an indication other than advanced melanoma.

### 2.3. Study Variables

We included patient demographics (i.e., age, gender) and clinical characteristics (i.e., melanoma stage III or IV, Eastern Cooperative Oncology Group (ECOG) score, presence of brain metastasis, lactate dehydrogenase (LDH) level, chronic steroid use, autoimmune diseases, and BRAF status) at the index date to describe the study population. The time-to-event endpoints that were computed for this study were the OS, rwPFS, rwTtNT, and rwToT. Additional secondary endpoints, the real-world tumor response rate (rwTRR) and the real-world tumor control rate (rwTCR), were also examined and are reported in the Appendix A. The definitions of the endpoints are presented in Table 1.

### 2.4. Statistical Analysis

The data were analyzed using the SAS v9.3 (SAS Institute Inc., Cary, NC, USA) and R v3.3 software packages. The descriptive statistics calculated included the frequencies and percentages of categorical variables, while the means, medians, standard deviations (SD), and ranges of continuous variables were determined for the demographic and clinical characteristics. The median follow-up times were estimated overall and by line of treatment using the reverse Kaplan–Meier estimator [31].

The time-to-event endpoints (OS, rwPFS, rwTtNT, and rwToT) were examined using the Kaplan–Meier (KM) method, and the OS and rwPFS rates at 6, 12, 18, and 24 months of follow-up were estimated. The OS and rwPFS were estimated in patients with at least 6 months of follow-up post-index or a survival and/or progression event if less than 6 months of follow-up. In addition, for the rwTtNT estimated rates for “not on subsequent treatment” are reported at 6, 12, 18, and 24 months. For rwToT, the restricted mean at 24, 30, and 36 months and estimates for being on treatment at 12, 24, 30, and 36 months of follow-up are reported. For rwTRR and rwTCR, the percentage response is reported. We report these endpoints for the overall sample in the main manuscript and separately by the first, second, and third+ lines of therapy in the Appendix A.

To investigate the relationship between the real-world endpoints, correlations of the rwTtNT, rwPFS, and rwToT to the OS were assessed at the patient level by estimating the Spearman rank correlation coefficients (0 = no association and 1 = perfect association) and the corresponding 95% CIs [23]. Spearman rank correlations are frequently used but do not take into account censoring, which can lead to biases in the estimated correlations. We, therefore, additionally applied a copula-based approach with an iterative multiple imputation method for the estimation of the correlation coefficients to account for censoring [32,33]. Correlations are reported for the overall sample and for the first line of therapy only, as there were only slight differences in terms of outcomes by therapy line.

## 3. Results

### 3.1. Patient Characteristics

Data were abstracted for 664 eligible patients (Table 2). The majority of patients were male (59.9%), and more than 90% of all enrolled patients had stage IV melanoma, assessed according to the 2018 (Eighth Edition) American Joint Committee on Cancer (AJCC) Melanoma staging criteria [34]. The median patient age was 70 years (range 22 to 96), and ECOG scores of 0 and 1 were reported for 40.8% and 21.4% of patients, respectively. Nearly a quarter (23.0%) of all included patients had a history of brain metastases. Furthermore, 39.0% of the patients had elevated LDH levels, with 6.5% of patients with LDH values greater than or equal to twice the upper limit of normal (ULN). Of those with available data, only 3.2% were on long-term steroid treatment, and 0.9% of patients had an autoimmune disease at the index date. A BRAF mutation was present in 32.4% of the patients. Sixty-four percent of the patients received pembrolizumab as the first line of therapy, 21.5% in the second line, and 18.1% in the third or higher lines of therapy. Regarding treatment history, 35.7% of the patients in the second line and 47.5 of patients in the third or higher lines were previously treated with BRAF/MEK inhibitors. Furthermore, prior treatment with ipilimumab was observed in 44.1% of patients in the second line and 60.8% of patients in the third or higher lines (Appendix A). The median (95% CI) follow-up time was 36.1 (33.5 to 38.3) months, 32.8 (29.4 to 38.3) months, 36.1 (32.8 to 40.2) months, and 46.4 (41.5 to 49.6) months, for overall, and the first, second, and third or higher lines of therapy, respectively.

### 3.2. Clinical Outcomes

The median (95% CI) OS for the overall sample was 30.5 (25.0 to 35.4) months, and the estimated probabilities of survival at 6, 12, 18, and 24 months after treatment initiation were 84.3%, 71.0%, 62.2%, and 55.4%, respectively (Appendix A). The median (95% CI) rwPFS for all included patients was 3.9 (3.5 to 4.9) months from the date of initiation of pembrolizumab. The estimated probability of remaining progression-free at 6, 12, 18, and 24 months was 42.8%, 30.7%, 24.5%, and 22.5%, respectively (Appendix A). The median rwTtNT was 10.7 months (95% CI, 9.0 to 12.9) for the overall study population treated with pembrolizumab. The estimates for rates of not on subsequent treatment were 65.5%, 47.6%, 37.5%, and 33.8% at 6, 12, 18, and 24 months (Appendix A). For OS, the rwPFS and rwTtNT outcomes were comparable for patients in the first and third+ therapy lines and slightly shorter in the second line compared to the first and third+ lines (e.g., first, third+, and second therapy line OS medians were 30.9, 33.5 and 26.5 months, respectively). 

Across the entire study population, the median rwToT was 6.2 months (95% CI, 5.1 to 6.8) (Appendix A), and the 2-year restricted mean rwToT was 9.3 months. The estimates for the restricted mean rwToT at 30 and 36 months were 9.9 and 10.4 months, respectively. The estimates for the restricted mean rwToT by line of therapy at 24 to 36 months ranged from 9.4 to 10.5 months for the first line, 8.2 to 9.2 months for the second line, and 10.2 to 11.5 months for the third+ lines of therapy.

The results for the secondary clinical endpoints of rwTRR and rwTCR are presented in the Appendix A. For the overall study population, the rwTRR (95% CI) was 24.4% (21.1 to 27.7), with 24.9% (20.7 to 29.2), 21.0% (14.3 to 27.7), and 26.7% (18.8 to 34.6) for the first, second and third+ lines of therapy, respectively. The rwTCR (95% CI) was 40.2% (36.5 to 43.9) for the overall study population and 43.4% (38.5 to 48.2), 35.0% (27.1 to 42.8), and 35.8% (27.3 to 44.4) for those in the first, second and third+ lines of therapy, respectively.

### 3.3. Correlations among Real-World Endpoints

#### 3.3.1. Spearman’s Rank Correlations

All real-world endpoints (rwTtNT, rwToT, rwPFS) showed significant correlations with the OS in the complete study population and the first line of therapy. Across the entire sample, the highest correlation (95% CI) with OS was present for rwTtNT (0.74 (0.70 to 0.78)) followed by rwToT (0.65 (0.61 to 0.70)). The rwPFS showed the lowest correlation with the OS (0.56 (0.50 to 0.61)) (Appendix A). Correlations were marginally lower for the first line of therapy (Appendix A) compared to the overall sample (see Table 3).

#### 3.3.2. Iterative Multiple Imputation (IMI) Correlation

All real-world endpoints (rwPFS, rwTtNT, rwToT) showed higher IMI correlations with the OS compared to those derived with the Spearman rank method. The highest correlation with OS emerged for rwTtNT (0.83 (0.79 to 0.86)), followed by rwToT (0.74 (0.69 to 0.79)), and rwPFS with the lowest correlation (0.69 (0.62 to 0.74)) (see Figure 1, Figure 2 and Figure 3). Correlations were only slightly lower for patients in the first line of therapy than for the overall sample (see Table 3).

Correlations between the OS and the rwPFS, rwTtNT, and rwToT for both methods in the overall study population and for patients who received pembrolizumab as first-line of therapy, along with their 95% CIs, are reported in Table 3.

## 4. Discussion

This study reported real-world clinical outcomes for advanced melanoma patients receiving pembrolizumab treatment and examined the associations between the OS and the rwPFS, rwTtNT, and rwToT in advanced melanoma patients treated with pembrolizumab in Germany to evaluate the outcomes and different surrogate endpoints for OS in real-world clinical practice. The median survival was 30.5 months for the overall patient sample. Consistently, the highest correlations were found between the OS and rwTtNT using both the IMI and Spearman rank-based methods, suggesting that rwTtNT may be a useful surrogate endpoint.

### 4.1. Clinical Outcomes

The median OS (30.5 months) and 2-year survival rate (55.4%) were found to be higher in this study compared to previous reports of the median OS, ranging from 19.4 to 30 months [16,18,20], and the 2-year survival rates (48% and 44%) reported by others [16,18]. A lower proportion of patients with a poor ECOG score in the present study may explain these differences from RWE studies with poorer OS outcomes. However, when compared to RCTs, the median OS in the present study is substantially shorter [10]. The more advanced illness of patients in our real-world sample (e.g., ECOG score >1, prevalence of brain metastases), some of whom would have been excluded from RCTs, could explain this difference. Moreover, the RWE sample was substantially older than the sample included in the RCT (median age 70 vs. 61).

The median rwPFS in the current study is comparable to the median rwPFS (3.9 vs. 4.2 months) reported by a recent US RWE study [18]. In addition, the result for the rwTtNT in the current study is similar to what has recently been reported for advanced melanoma of cutaneous origin patients in a US sample (10.7 vs. 11.2 months) [16]. The median rwToT of 6.2 months (2-year restricted mean rwToT of 9.3 months) in the current study is slightly longer than the previously reported rwToT by others (median rwToT of 4.9 months, 2-year restricted mean treatment duration of 8.0 months) [16].

The clinical outcomes identified in this sample appear to be similar among lines of therapy, which may be due to the relatively short length of follow-up in some patients. An update of the clinical outcomes will be completed in a larger sample with longer follow-up in the near future.

### 4.2. Evaluation of Relationships among Real-World Outcome Endpoints

In real-world practice, capturing outcomes such as OS and rwPFS can be challenging due to, for example, recording mechanisms or treatment that continues beyond progression to preserve tumor control. For the purpose of this study, we have, therefore, evaluated the correlation between different possible surrogate endpoints and the OS in alignment with the FoCR group’s framework to evaluate real-world endpoints [27]. The FoCR framework used Spearman’s rank correlation coefficient, which is among the most often used statistical methods in medical research. However, it ignores censoring, thus leading to biased estimates, particularly if censoring is prevalent. We, therefore, additionally used the IMI method, which iteratively augments normal deviates related to censored times [32,33]. We found high correlations between all the investigated surrogate endpoints and OS. Correlations were higher with the copula-based IMI than the more widely used Spearman rank method, indicating that accounting for censoring using the IMI-based method may affect correlation estimates. The rwTtNT had a higher correlation with OS (rIMI = 0.83) than the rwPFS (rIMI = 0.69) and rwToT (rIMI = 0.74). The magnitude of the correlations differed only slightly by line of therapy. The findings suggest that rwTtNT as defined (initiation of next line of therapy or death) in this study is a reliable reflection of OS and clinical benefit in this setting, supporting its application as a clinically meaningful surrogate endpoint. Using the rwTtNT as the endpoint has additional advantages. It can be estimated with shorter follow-up than OS and avoids challenges associated with pin-pointing the exact date of progression. It also offers an assessment of the duration of therapeutic benefit as well as a better reflection of the patients’ treatment experiences than more conventional endpoints, such as rwPFS [28].

### 4.3. Strengths and Limitations of the Study

Our study also provides novel insights into the surrogate endpoints for OS and the outcomes observed in pembrolizumab-treated patients with advanced melanoma in Germany. We used a copula-based approach that accounts for censoring, thus yielding more robust results than the more widely used Spearman rank correlations. The well-maintained ADOReg database allowed for the inclusion of a large sample of a heterogeneous patient population. Patients with a poor ECOG score (>1), primary melanoma of other than cutaneous origin, and co-morbidities, who are generally excluded or under-represented in RCTs, were included in this study, making it possible to present a more comprehensive picture of real-world clinical outcomes. Most patients were male and elderly and had an ECOG score between 0 and 1. Almost one-third of all evaluated melanomas were positive for a BRAF mutation. The demographic and clinical characteristics of patients in this study are in line with earlier published real-world advanced melanoma studies [16,18,19].

While novel and important, this study’s findings need to be interpreted in light of several important methodological considerations. First, while the ADOReg database is the largest and most complete melanoma registry in Germany, the data for this observational retrospective study was not primarily recorded for research purposes but based on physician reports from clinical practice. As a result, the dataset was not always complete, with missing data on real-world outcomes for some individuals (e.g., ~9% for OS and ~6% for rwPFS). Second, the generalizability of the results beyond Germany and those being treated with immunotherapy may be limited due to differing prescribing practices, approaches to routine treatment, and treatment effects. Additional prospective investigations of pembrolizumab and advanced melanoma real-world outcomes are, therefore, needed to corroborate and extend our findings.

## 5. Conclusions

Our study yielded important new insights into the outcomes of patients with advanced melanoma and surrogate endpoints for OS for real-world studies. Our findings support the effectiveness of pembrolizumab in patients with advanced melanoma and that all investigated real-world endpoints are suitable surrogates for OS. However, rwTtNT had the highest correlation with OS information, suggesting it as the most reliable indicator of OS among the investigated endpoints.

## Figures and Tables

**Figure 1 cancers-14-01804-f001:**
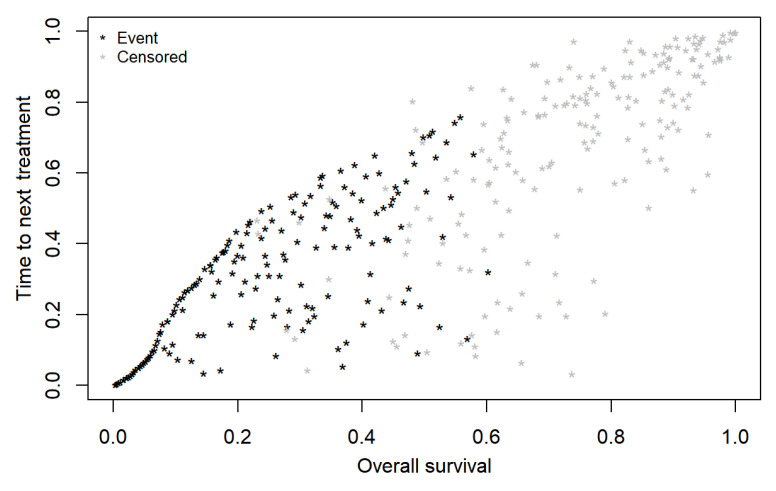
Bivariate copula plot depicting correlation between real-world overall survival (rwOS) and real-world time to next treatment (rwTtNT) of patients with advanced melanoma treated with first-line pembrolizumab.

**Figure 2 cancers-14-01804-f002:**
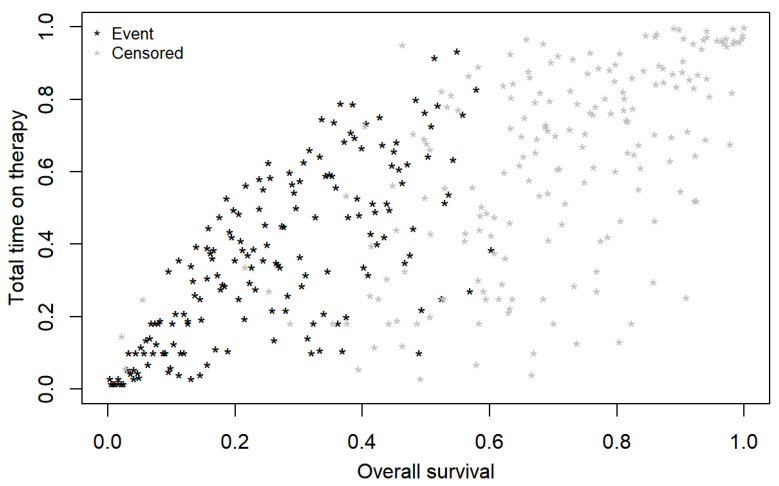
Bivariate copula plot depicting correlation between real-world overall survival (rwOS) and real-world time to next treatment (rwToT) of patients with advanced melanoma treated with first-line pembrolizumab.

**Figure 3 cancers-14-01804-f003:**
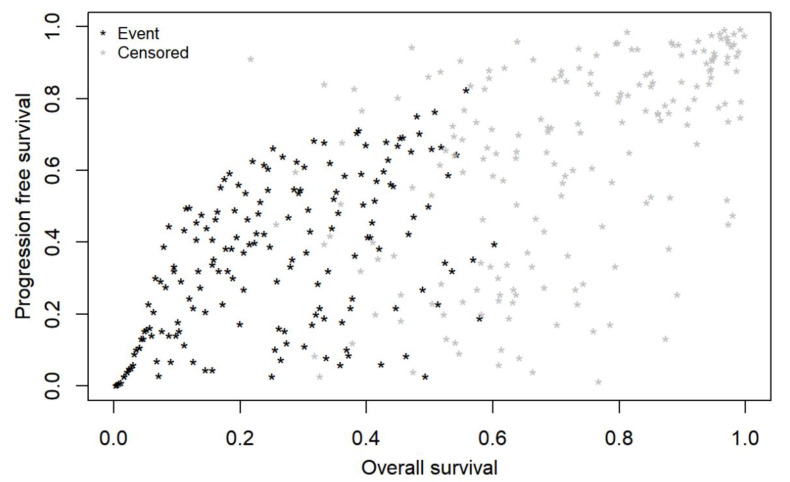
Bivariate copula plot depicting correlation between real-world overall survival (rwOS) and real-world progression-free survival (rwPFS) of patients with advanced melanoma treated with first-line pembrolizumab.

**Table 1 cancers-14-01804-t001:** Definition of real-world endpoints used in this study.

Endpoint	Definition
* **Primary** *	
Overall survival (OS)	The time interval from index date to date of death. Patients alive at the date of last contact were censored.
Real-world progression-free survival (rwPFS)	The time interval from index date to physician-reported date of progression, death date or start date of a new treatment due to progression of disease (whichever came first). Patients without a progression event or date of death were censored at the date of last contact.
Real-world time to next treatment (rwTtNT)	The time interval between index date and date of (i) start date of a new treatment/regimen or (ii) death. Patients who did not initiate new treatment nor died were censored at the date of last recorded encounter in the database.
Real-world time on treatment (rwToT)	The time interval between index date and the date of last dose of pembrolizumab within the same line of therapy (last dose date minus first dose date +1 day) at or before decision to discontinue * treatment or date of death if the patient died during treatment. Patients with ongoing pembrolizumab treatment or lost to follow-up were censored at the date of last contact.
* **Secondary** *	
Real-world tumor response rate (rwTRR)	The proportion of patients with a complete response or partial response based on real-world response assessments^#^ relative to all patients initiating treatment. (The best therapy response using both the clinical assessments in the medical record and the radiological assessment in the staging findings are captured within the ADOReg database).
Real-world tumor control rate (rwTCR)	The proportion of patients who had a complete response, partial response, or stable disease based on real-world response assessments ^†^. (The best therapy response using both the clinical assessments in the medical record and the radiological assessment in the staging findings are captured within the ADOReg database).

Note. * Complete discontinuation refers to a treatment discontinuation for at least 120 days or if subsequent therapy line was initiated. ^†^ Complete response: complete resolution of all visible disease; partial response: disease still present, with partial reduction in size of visible disease in some or all areas without any areas of increase in visible disease; stable disease: no change in overall size of visible disease or mixed response.

**Table 2 cancers-14-01804-t002:** Baseline characteristics of patients (*N* = 664) at the index date (initiation of pembrolizumab treatment).

Characteristic	
**Age (years)**	
Mean (SD)	67.4 (13.2)
Median (min–max)	70 (22–96)
**Gender, *n* (%)**	
Male	398 (59.9)
Female	266 (40.1)
**Stage (2018 AJCC Melanoma** **Staging), *n* (%)**	
Stage III	62 (9.3)
Stage IV	602 (90.7)
**Origin of primary melanoma, *n* (%)**	
Cutaneous	537 (80.9)
Mucosal	17 (2.6)
Ocular	30 (4.5)
Unknown primary	80 (12.0)
**ECOG score, *n* (%)**	
0	271 (40.8)
1	142 (21.4)
2	30 (4.5)
3	7 (1.1)
Missing	214 (32.2)
**Line of therapy, *n* (%)**	
1st	401 (60.4)
2nd	143 (21.5)
3rd+	120 (18.1)
**Brain metastasis, *n* (%)**	
Present	154 (23.2)
Absent	510 (76.8)
**LDH level, *n* (%)**	
WNL	402 (60.5)
>1-2X ULN	216 (32.5)
>2X ULN	43 (6.5)
Missing	3 (0.5)
**Chronic Steroid use, *n* (%)**	
Yes	21 (3.2)
**History of autoimmune disease** **at index, *n* (%)**	
Yes	6 (0.9)
**BRAF status, *n* (%)**	
Wildtype (negative)	360 (54.2)
Positive	215 (32.4)
Missing/Unknown	89 (13.4)

Note. AJCC: American Joint Committee on Cancer; ECOG: Eastern Cooperative Oncology Group; LDH: lactate dehydrogenase; SD: standard deviation; ULN: upper limit of normal; WNL: within normal limit.

**Table 3 cancers-14-01804-t003:** Correlations among real-world outcome endpoints.

Comparison	Spearman’s Rank Correlation (95% CI)		IMI Correlation (95% CI)	
	Overall	1st Line	Overall	1st Line
**OS vs. rwPFS**	0.56 (0.50, 0.61)	0.52 (0.44, 0.59)	0.69 (0.62, 0.74)	0.68 (0.59, 0.74)
**OS vs. rwTtNT**	0.74 (0.70, 0.78)	0.70 (0.65, 0.75)	0.83 (0.79, 0.86)	0.83 (0.77, 0.87)
**OS vs. rwToT**	0.65 (0.61, 0.70)	0.62 (0.55, 0.68)	0.74 (0.69, 0.79)	0.75 (0.68, 0.80)

Note. IMI: iterative multiple imputation; CI: confidence interval; OS: real-world overall survival; rwPFS: real-world progression; rwToT: real-world time on treatment; rwTtNT: real-world time to next treatment.

## Data Availability

Data is not publicly available and cannot be shared.

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
