# Peer review of "Real-World Therapy with Pembrolizumab: Outcomes and Surrogate Endpoints for Predicting Survival in Advanced Melanoma Patients in Germany"

_cancers, 2022, doi:10.3390/cancers14071804_

Round 1
Reviewer 1 Report
Authors reported the real-word experience of a large population of 664 patients treated with pembrolizumab for advanced melanoma in Germany to evaluate outcomes and different surrogate endpoints for OS.
They found the highest correlations between OS and real-world time to next treatment (rwTtNT) using both IMI and Spearman rank based methods, suggesting that this parameter could be a useful surrogate endpoint for OS in real-worf studies.
The study is well designed and written, results are presented properly, and the conclusions drawn are supported by the data provided.
Globally, the study confirms on a large real-life patient population previously published data.
Minor points
Why Authors reported pembrolizumab only experience (what about other antiPD-1 as nivolumab?
Regarding patients treated with pembro as second/third line (about 40% of patients), what about previous therapies used?
How was the population with the BRAF mutation (32.4% of patients) managed with regards to targeted therapy?
Author Response
The authors thank the reviewers’ comments and appreciation of our manuscript. As we agree with the suggested improvements of the manuscript, changes were performed accordingly throughout the manuscript.
Reviewer 1
Authors reported the real-word experience of a large population of 664 patients treated with pembrolizumab for advanced melanoma in Germany to evaluate outcomes and different surrogate endpoints for OS.
They found the highest correlations between OS and real-world time to next treatment (rwTtNT) using both IMI and Spearman rank based methods, suggesting that this parameter could be a useful surrogate endpoint for OS in real-worf studies.
The study is well designed and written, results are presented properly, and the conclusions drawn are supported by the data provided.
Globally, the study confirms on a large real-life patient population previously published data.
Answer: We would like to thank for the appreciation of our manuscript.
Minor points
- Why Authors reported pembrolizumab only experience (what about other antiPD-1 as nivolumab)?
Answer: We acknowledge that it would be relevant reporting on other anti-PD-1 therapies. However, in this study, we were only interested in analyzing pembrolizumab and in providing insights about its outcomes in advanced melanoma patients.
- Regarding patients treated with pembro as second/third line (about 40% of patients), what about previous therapies used?
Answer: To address this question, we have added a table (new Table 1) to the Supplementary Material and added a respective sentence to the results section. Patients in second line or higher had primarily received BRAF/MEK inhibitors or anti-CTLA4 therapy in previous lines.
- How was the population with the BRAF mutation (32.4% of patients) managed with regards to targeted therapy?
Answer: Thank you for your question. Although the reviewer raised a relevant point, in this study, we focused only on pembrolizumab. Regarding the treatment history of the current study population, 35.7% of patients in second line and 47.5 of patients in third line or higher where previously treated with BRAF/MEK inhibitors (results presented in new Table 1 in Supplementary Material). Further studies could be conducted with regard to targeted therapy, and to specify the treatment patterns for this group of patients.
Reviewer 2 Report
The authors summarize real-world data obtained from 35 geographically diverse skin cancer centers or practice-based dermato-oncologists certified by the German Cancer Society. The study included 664 eligible patients data. Majority of patients tumour were in stage IV (602 patients), 154 with brain metastasis and 32.4 % were detected for BRAF mutation. From the clinical point of view, it is a very important and interesting study.
The study highlights important new insights into outcomes of patients with advanced 369 melanoma and surrogate endpoints for overall survival for real-world studies and the published data support the effectiveness of pembrolizumab in real-world clinical practice.
The study is well designed, the data are well summarized. Appropriate statistical analysis were performed.
Author Response
The authors thank the reviewers’ comments and appreciation of our manuscript. As we agree with the suggested improvements of the manuscript, changes were performed accordingly throughout the manuscript.
The authors summarize real-world data obtained from 35 geographically diverse skin cancer centers or practice-based dermato-oncologists certified by the German Cancer Society. The study included 664 eligible patients data. Majority of patients tumour were in stage IV (602 patients), 154 with brain metastasis and 32.4 % were detected for BRAF mutation. From the clinical point of view, it is a very important and interesting study.
The study highlights important new insights into outcomes of patients with advanced 369 melanoma and surrogate endpoints for overall survival for real-world studies and the published data support the effectiveness of pembrolizumab in real-world clinical practice.
The study is well designed, the data are well summarized. Appropriate statistical analysis were performed.
Answer: We would like to thank the overall appreciation of the manuscript and the reviewer for his/her feedback.
Reviewer 3 Report
This is an interesting manuscript in advanced melanoma treatment addressing the important topic of follow up for patients in a real world setting. There is an absolute need for new approaches on the understanding how we best analyze real world efficacy data where the tools we use in prospective clinical trials for obvious reasons is not applicable or at least are associated with large uncertainties for RWD. Therefore it is of special interest to see the analysis of the surrogate endpoint ”real-world time-to-next-treatment” which seems to be closely linked with overall survival. An important observation of potential value in similar studies in the future.
Specific comments
- The use of monotherapy pembrolizumab in Germany needs to be presented in a relation to other therapies for the reader. What group of patients receive(d) PD-1 inhibitors in monotherapy during this period, who receives combination therapy, i.e. ipi-nivo, and who receives BRAF-MEK inhibition and in what sequence of therapy 1L, 2L, 3L? Altough the main objectives of this study were to examine clinical outcomes this data needs to be included in the ms to better understand the results and the conclusions made.
- Subgropus of patient characteristics needs a more detailed presentation, stage III, M1a, M1b, M1c, M1d (AJCC 8th)? Is the results obtained valid for all subgroups of advanced cutaneous melanoma? At page 10, l 150 its stated that also other types of melanoma than cutaneous were included. Please give data for the number of ocular, mucosal, unknown primary etc in table 2 The relatively low number of BRAF mutations might be explained by missing data or a large proportion of non-cutaneous melanoma?
- ECOG status has been shown to be a very important prognostic factor for survival in RW studies. ECOG status is missing in 32% of the patients and an ECOG 2-3 only in 5.6% of the study population. Are there any difference in results between the ECOG 0-1 group vs the other 2-3/missing groups?
- 2% of the patients had a history of brain mets at start of therapy. How many developed brain mets overtime and did that influence the results?
- In the definitions for response in the database both clinical and radiological response are used as variables. What defines which one of the two is superior over the other?
- Considering the similar results for clinical outcomes at different lines of therapy the short follow up time ia mentioned as one potential reason. Its not fully clear in the text which follow up times were observed for different subgroups, i.e. lines of therapy 1st, 2nd, 3d. What was minimum and medium follow up time?
- Table 1 is hard to read, please revise layout (journal editing?)
Author Response
The authors thank the reviewers’ comments and appreciation of our manuscript. As we agree with the suggested improvements of the manuscript, changes were performed accordingly throughout the manuscript.
This is an interesting manuscript in advanced melanoma treatment addressing the important topic of follow up for patients in a real world setting. There is an absolute need for new approaches on the understanding how we best analyze real world efficacy data where the tools we use in prospective clinical trials for obvious reasons is not applicable or at least are associated with large uncertainties for RWD. Therefore it is of special interest to see the analysis of the surrogate endpoint ”real-world time-to-next-treatment” which seems to be closely linked with overall survival. An important observation of potential value in similar studies in the future.
Answer: The authors thank the reviewer for the comments and appreciation of our manuscript and the opportunity to improve it. The reviewer’ comments were considered and a detailed response to each one can be found below.
Specific comments
- The use of monotherapy pembrolizumab in Germany needs to be presented in a relation to other therapies for the reader. What group of patients receive(d) PD-1 inhibitors in monotherapy during this period, who receives combination therapy, i.e. ipi-nivo, and who receives BRAF-MEK inhibition and in what sequence of therapy 1L, 2L, 3L? Altough the main objectives of this study were to examine clinical outcomes this data needs to be included in the ms to better understand the results and the conclusions made.
Answer: Thank you for your comment. We acknowledge the relevance of the points mentioned by the reviewer. However, this study only focused on pembrolizumab. Regarding the treatment history of the current study population, 35.7% of patients in second line and 47.5 of patients in third line or higher where previously treated with BRAF/MEK inhibitors (results presented in new Table 1 in Supplementary Material). Further studies could be conducted with regard to targeted therapy, and to specify the treatment patterns for this group of patients.
- Subgropus of patient characteristics needs a more detailed presentation, stage III, M1a, M1b, M1c, M1d (AJCC 8th)? Is the results obtained valid for all subgroups of advanced cutaneous melanoma? At page 10, l 150 its stated that also other types of melanoma than cutaneous were included. Please give data for the number of ocular, mucosal, unknown primary etc in table 2. The relatively low number of BRAF mutations might be explained by missing data or a large proportion of non-cutaneous melanoma?
Answer: The authors acknowledge the reviewer’s comment. We now provide information about the type of melanoma in Table 2 (please see updated manuscript file). More specific information on M-status at primary diagnosis is unfortunately not available at this point. The relatively low proportion of patients with BRAF mutated tumors may be explained by the fact that BRAF/MEK therapy is not a viable option for patients with BRAF wildtype tumors, but pembrolizumab - the therapy in focus for this study – is a treatment option for these patients.
- ECOG status has been shown to be a very important prognostic factor for survival in RW studies. ECOG status is missing in 32% of the patients and an ECOG 2-3 only in 5.6% of the study population. Are there any difference in results between the ECOG 0-1 group vs the other 2-3/missing groups?
Answer: We recognize that the ECOG status is an important prognostic factor, and that a stratified analysis would be relevant. In a previous analysis, it was observed that melanoma patients with missing ECOG had similar outcomes as patients with an ECOG of 0-1 (Moser JC et al. 2019; https://doi.org/10.1002/cam4.2625). The sample size of the current study does not allow to provide meaningful results for the relatively small sub-group of patients with an ECOG of 2-3.
- 2% of the patients had a history of brain mets at start of therapy. How many developed brain mets overtime and did that influence the results?
Answer: This is an important consideration for melanoma treatment. However, this topic is planned to be investigated in another study based on ADOREG data.
- In the definitions for response in the database both clinical and radiological response are used as variables. What defines which one of the two is superior over the other?
Answer: We would like to clarify, that the documentation guidelines of ADOREG state that both the clinical assessment and the radiological assessment should be considered when documenting best overall response. The investigator decides how to rate the response and what is entered into the ADOREG registry as best overall response. The analysis did not apply a separate definition to decide best overall response.
- Considering the similar results for clinical outcomes at different lines of therapy the short follow up time ia mentioned as one potential reason. Its not fully clear in the text which follow up times were observed for different subgroups, i.e. lines of therapy 1st, 2nd, 3d. What was minimum and medium follow up time?
Answer: We added information on follow-up times observed for different subgroups based on line of therapy in the results section of the manuscript. The reverse Kaplan-Meier estimator was used to estimate median follow-up times from index date to last contact date, censoring patients who died during follow-up. Minimum follow-up was 0 days for each of the subgroups by treatment line. The median follow-up times stratified by line of therapy where similar except for patients receiving 3rd or higher lines of therapy, for which an approximately 10-month longer median follow-up was observed. The median (95% CI) follow-up time for the overall sample was 36.1 (33.5 to 38.3) months, whereas for patients at 3rd or higher lines of therapy it was 46.4 (41.5 to 49.6) months. The longer follow-up time may be a chance finding, as censoring patterns were similar in all three subgroups.
- Table 1 is hard to read, please revise layout (journal editing?)
Answer: The author thanks the reviewer suggestion. We performed changed in the table to improve its reading.